**Data Availability Statement:** All relevant data are within the paper and its Supporting information files.

# Pooled prevalence and its determinants of stunting among children during their critical period in Ethiopia: A systematic review and meta-analysis

**Amare Kassaw** [ORCID][1]*, **Yohannes Tesfahun Kassie**[2], **Demewoz Kefale**[1], **Molla Azmeraw**[3], **Getachew Arage**[1], **Worku Necho Asferi**[4], **Tigabu Munye**[5], **Solomon Demis**[4], **Amare simegn**[6], **Muluken Chanie Agimas**[7], **Shegaw Zeleke**[5]

1 Department of Pediatrics and Child Health Nursing, College of Health Sciences, Debre Tabor University, Debre Tabor, Ethiopia, 2 Department of Emergency and Critical Care Nursing, College of Health Sciences, Debre Tabor University, Debre Tabor, Ethiopia, 3 Department of Pediatrics and Child Health Nursing, College of Health Sciences, Woldia University, Woldia, Ethiopia, 4 Department of Maternal and Neonatal Health Nursing, College of Health Sciences, Debre Tabor University, Debre Tabor, Ethiopia, 5 Department of Adult Health Nursing, College of Health Sciences, Debre Tabor University, Debre Tabor, Ethiopia, 6 Department of Midwifery, College of Health Sciences, Debre Tabor University, Debre Tabor, Ethiopia, 7 Department of Epidemiology and Biostatics, Institute of Public Health, College of Medicine and Health Sciences, University of Gondar, Gondar, Ethiopia

* amarekassaw2009@gmail.com

## Abstract

### Background

Stunting is a major public health concern, particularly in low and middle-income countries. Globally, nearly 149 million under-five children are suffering from stunting. Despite it can occur in all age groups, the impact is more severe among children age less than 24 months as this period is critical time of very rapid growth and development. Therefore, this review aimed to determine the pooled prevalence and determinants of stunting among children during this critical period in Ethiopia.

### Methods

The literature search was conducted using international electronic data bases (pumed, Google scholar, CINHAL, Hinari, open Google) and the hand search of reference lists of eligible articles. The presence of heterogeneity between studies was evaluated using Cochrane Q-test and $I^2$ test statistics and sensitivity analysis was also checked. Small study effect was checked through graphical and statistical test. Sub-group analysis was performed to handle heterogeneity.

### Results

This study included 14 studies with a total sample size of 8,056 children. The overall pooled estimate of stunting was 35.01(95% CI: 24.73–45.28, $I^2$ = 98.98%) in the country with the highest prevalence in Amhara region. Increased Child's age (OR = 3.83; 95% CI: 2.47–

**Funding:** The author(s) received no specific funding for this work.

**Competing interests:** The authors have declared that no competing interests exist.

5.18, $I^2$ = 97.76%), no maternal education (OR = 2.90; 95%CI: 1.59–4.20, $I^2$ = 89.73%), no maternal postnatal follow up (OR = 1.81; 95% CI:1.51–2.10) less than four food diversity of the child (OR = 2.24;95%CI; 1.94–2.55,$I^2$ = 21.55%), low maternal body mass index, failure to colostrum and exclusive breast feeding, two and more under five children in the household and poor wealth index of the family were significant factors of stunting.

## Conclusion and recommendations

The pooled prevalence of stunting among children during their critical time is high. Increased Child's age, no maternal education and no maternal postnatal follow up, less than four food diversity of the child, low maternal body mass index, failure to colostrum and exclusive breast feeding, two and more under five children in the household and poor wealth index of the family were determinants of stunting. Therefore, providing continuous maternal postnatal follow up, increase awareness of mothers on importance of colostrum and exclusive breast feeding, feeding of children the recommended variety of foods and at large to improve the wealth status of the households are crucial interventions to meet national and international targets of zero stunting in children less than 2 years.

## Introduction

Childhood undernutrition mainly expressed in the form of stunting, wasting, underweight and deficiencies of micronutrients [1]. It is a major public health concern, particularly in low and middle-income countries [2]. Stunting is a measure of chronic undernutrition and it is defined as the percentage of children whose height for age is below minus two standard deviations from the median of the World Health Organization (WHO) [3].

Despite stunting can occur in all age groups, the impact is more severe among children age less than 24 months as this period is critical time of very rapid growth and development and in high demands for nutrients [4]. Evidences indicated that the period from pregnancy through the first 2 years of life (known as the first 1,000 days) is a critical window of opportunity for the prevention of malnutrition [5, 6]. This period represents a life window when growth rates and neuroplasticity are at their peak and where nutritional deficiencies can exert their most devastating impacts [7]. The window of opportunity to prevent undernutrition ends at 2 years of age and it becomes increasingly difficult to reverse growth faltering and prevent stunting after this critical period [8]. Stunting results in increased child morbidity, diminished cognitive and physical development, higher susceptibility to chronic diseases in adulthood, increased risk of degenerative diseases, reduced productive capacity and poor health, poor school performance and higher risk of mortality [9, 10].

Globally, nearly 149 million under-five children are suffering from stunting. Among theses the highest prevalence is found in South Asia (34.4%) followed by Eastern and Southern Africa 33.6% and the lowest share is found in North America, 2.6%. In South Asia and Sub-Saharan Africa, 1 among 3 children under- five is stunted [11]. The number of stunted children is still high in Africa regions like Western Africa (31.4%), middle Africa (32.5%) and Eastern Africa (36.7%) [12]. In Ethiopia according to 2019 EDHS, 37% of children under-five are stunted and the prevalence of stunting generally increases steadily with age (from 22% among children 6–8 months up to 44% of children 48–59 months) [13].

Several studies indicated that various factors contribute to the developments of stunting in less than 24 months children. For example, maternal age [14, 15], maternal education [16],

child age [17], having more than one child under 2 years of age in the household [18], minimum dietary diversity, consumption of animal sourced food, child's sex [19], diarrhea, respiratory infection [20], lower wealth quintile [21], household dietary diversity, early initiation of complementary feeding [22] are some of the associated factors.

Worldwide, there is a target to reduce stunting from 21.9% in 2018 to 12.2% by 2030 [11], including the target of reducing the number of stunted children under the age of five by 40% at the end of 2025 which is adopted by World Health Organization (WHO) [12]. The target will be achieved when the focus of prevention efforts center around the critical period (the first 1000 days) of life, because this is when nutrition interventions have been proven to offer children the best chance to survive and reach optimal growth and development [23]. Similarly, Ethiopia has planned and working through different interventions like National Nutrition Program (NNP) [24] and Seqota Declaration to reduces stunting to 26% by 2020 [25] and zero stunting in children under 24 months old by 2030 [26]. However, 2023 national food and nutrition strategy baseline survey disclosed that stunting still remains a major public health problem in which the prevalence is 39% in under-five children [27].

In Ethiopia various studies [4, 21, 22, 28–36] showed that the prevalence of stunting under 24 months have great inconsistence thought the nation ranging from 15.7% [32] in Oromia region to 71.8% [4] in South Nation and Nationalities and Peoples Regional States(SNNPRS). This inconsistence initiates nationally pooled evidence about the prevalence of stunting among children age less than 24 months /critical period. Moreover, the determinants of stunting have been also reported inconclusively.

Therefore, the aim of this systematic review and meta-analysis was to determine the pooled national prevalence of stunting among children less than 24 months and its associated factors. Evidence from this study will be used to as one input to achieve the target of zero stunting by 2030 held by SDG and Seqota declaration in Ethiopia.

## Methods

### Search strategy and PROSPERO registration

The included studies were selected using international electronic data base (pumed, Google scholar, CINHAL, Hinari) open Google and the hand search of reference lists of eligible articles were also searched and investigated. Furthermore, unpublished studies were also reviewed out from research centers and library sources. The searches were restricted to full text articles, human studies and English language studies. The search was included the following search terms/keywords and Mesh terms: "Prevalence", "proportion", "burden", "magnitude", "associated factors", "Risk Factors" "determinants", "predictors", "stunting", "chronic malnutrition", "Nutritional Status", "children 6–23 months", "children less than 2 years", and "Ethiopia". Using these key terms the search map was built: "Prevalence" [37] OR proportion OR burden OR magnitude OR epidemiology AND associated factors OR risk factors [37] OR determinants OR etiology OR predictors AND Stunting OR chronic malnutrition OR chronic undernutrition OR undernutrition OR nutritional Status [37] AND children 6–23 months OR under two years OR infants and young child OR children less than 2 years OR children 6–24 months AND Ethiopia on pumed data base (S1 Table).

To access eligible articles from electronic data bases, we used an adapted PICO/PEO mnemonic principle.

- **P**articipants/populations: children aged less than two years and whose height/age < -2Zscore.

- **I**ntervention/exposure group: under two years children with stunting.

- **C**omparison/comparator: well-nourished children

- **O**utcome of interest: prevalence of stunting among children less than two years.

This systematic review and meta-analysis was registered at the Prospero with a registration number of CRD42023414789 (https://www.crd.york.ac.uk/prospero/#myprospero).

## Inclusion and exclusion criteria

Both published and unpublished cross-sectional and case control studies that report the prevalence and determinants of stunting among children less than 24 months in Ethiopian were included. The studies were selected if their publication period is between 2013/1/1 and 2023/4/27 G.C. On the other hand, articles with no abstracts, case series, case reports, qualitative studies and studies with no report of prevalence/associated factors were excluded from this study.

## Outcome measurement

This review has intended to determine two main outcomes. The first outcome is pooled prevalence of stunting among children less than 24 months old and the second is its determinants. Extracted variables from each single study considered as an independent factor for stunting. Standard error and Odds ratio were calculated to determine the effect measures of prevalence and determinants respectively.

## Operational definition

**Critical period.**  The period from pregnancy through the first 2 years of life (known as the first 1,000 days) is a critical window of opportunity for the prevention of malnutrition. The window of opportunity to prevent undernutrition ends at 2 years of age and it becomes increasingly difficult to reverse growth faltering and prevent stunting after this critical period [5, 8].

## Study selection and screening process

Studies reviewed through different electronic data bases were screened by two independent authors (AK and SZ) to establish potentially relevant articles. The extracted studies were exported to endnote X8 software and then duplicate articles were removed whereas full-text articles were downloaded. Any disagreements between two authors were solved through discussion and other reviewer members (MA, SD & DK).

## Data extraction

After screened eligible studies from different electronic data bases, the relevant data were extracted by two authors (AK& WN). Any discrepancy between two authors were handled by discussion and other invited reviewers (AS&TM). For each included study, authors' name, publication year, study region, study design, study setting, sample size, response rate, standard error and prevalence of stunting were extracted on microsoft excel spread sheet. During critical appraisal of each primary study, more emphasis was given to the appropriateness of the study objectives, study design, sampling technique, data collection technique, statistical analysis, any sources of bias and its management technique.

## Quality assessment

The Joanna Briggs Institute (JBI) critical appraisal checklists for observational studies were used to determine the quality of the original studies [38]. Based on this, all the eligible studies

were critically appraised by two independent reviewers (YT and GA) and scored for the validity of their results. Dissensions between reviewers were settled by a free discussion. Accordingly, among the 12 cross- sectional studies, 7 studies were scored seven of eight questions 87.5%(low risk). Whereas, 5 of them scored 7 out of eight questions 75%(low risk). Similarly both of two case—control studies were appraised and scored greater than 80% (low risk) (S2 Checklist).

## Statistical analysis

The extracted data were exported to STATA version17 software for analysis. The pooled prevalence of stunting among children less than 24 months and its determinants were estimated by random effect model using DerSimonian-Laird model weight [39]. The presence of heterogeneity between studies was evaluated using Cochrane Q-test and the $I^2$ statistics [40]. Based on the statistical test result, there is significant heterogeneity between studies ($I^2$ = 98.98, p-value<0.001). To adjust random variation of estimated points between original studies, a sub group analysis was carried out by study regions, sample size and residence. Sensitivity analysis was also done to examine the effect of a single study on the pooled estimate. Moreover, small study effect was checked through graphical (funnel plot) and statistical (Egger's) test [41].

## Ethical consideration

Ethical clearance is not needed for this Systematic Review and Meta-Analysis.

## Results

### Study selection and identification

A total of 1013 papers were found from different electronic databases and other approach of searching. Among these, a total of 658 duplicate articles were excluded. After reading the title and abstract, 325 articles were removed as they are not relevant for this systematic review. Again, 13 articles were excluded due to poor quality and outcome not well defined. Finally, 14 articles were eligible for the final systematic review and meta-analysis. From these, only 12 studies were used for estimating the pooled prevalence of stunting (Fig 1).

### Characteristics of included studies

This systematic review and meta-analysis included 14 studies from different regions of Ethiopia with a total sample size of 8,056 children. The prevalence of stunting among the eligible studies varied from 15.7% [32] to 71.8% [4]. From included studies,12 of them were conducted in cross-sectional study [4, 21, 22, 28–36] while two were employed case control study design [42, 43]. Regarding the study region, three studies from Amhara region [31, 33, 36], three from Oromia region [28, 32, 34] and other three studies from SNNPRS [4, 22, 29], one from Addis Ababa [35], three from other regions of the country [30, 42, 43] and one national study in Ethiopia [21] were included (Table 1).

### The pooled prevalence of stunting among children during critical period in Ethiopia

Of all 14 studies, only 12 articles were used for estimating the pooled prevalence of stunting among children in critical period. Using random effects model, the overall pooled estimate of stunting was 35.01(95% CI: 24.73–45.28) with significant heterogeneity between studies ($I^2$ = 98.98, P-value<0.001) (Fig 2).

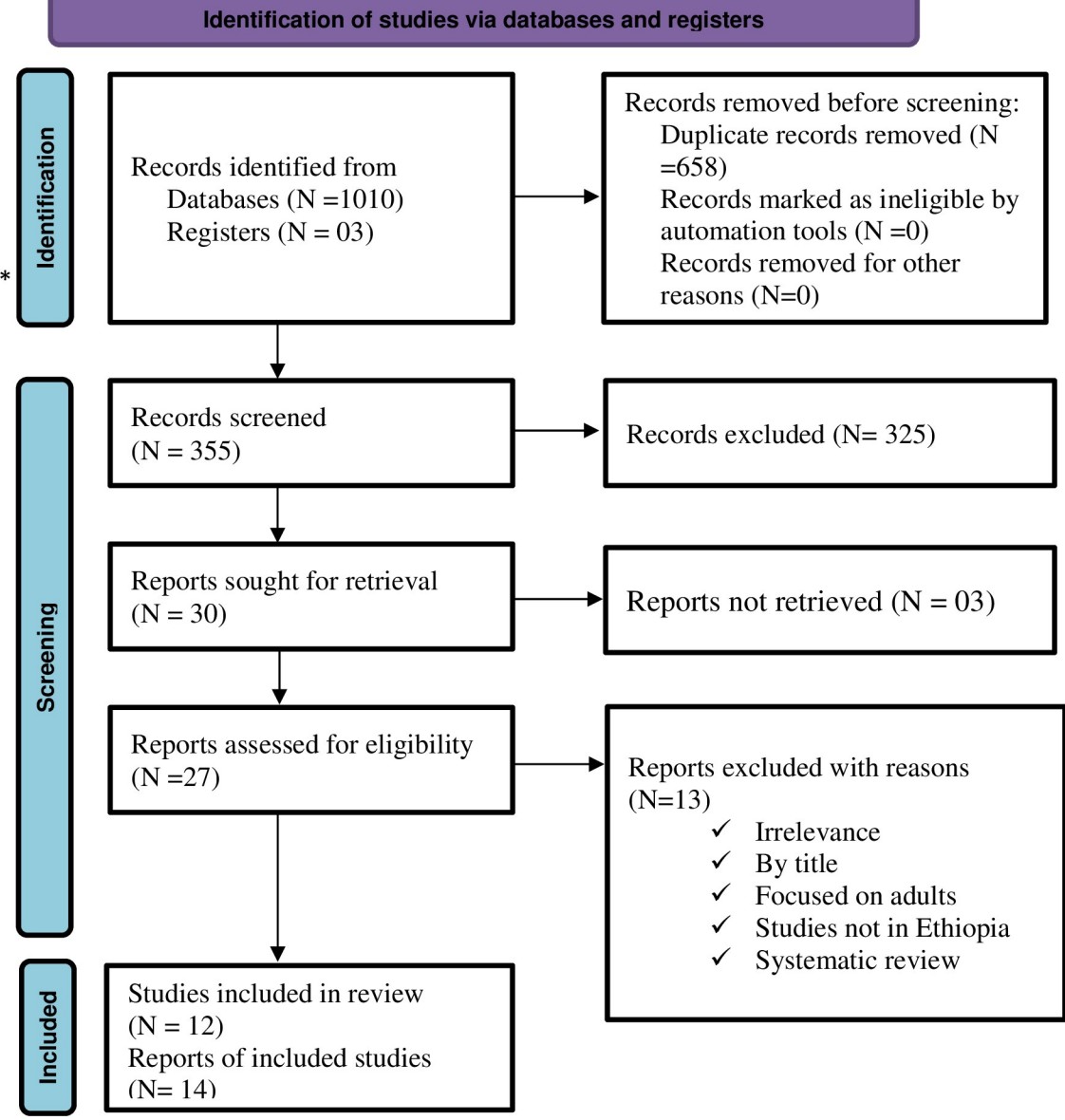

**Fig 1. PRISMA flow diagram of article selection for systematic review and meta-analysis of the prevalence of stunting among children during critical period and its determinants in Ethiopia.**

## Publication bias

The Egger's statistical test showed that there is no evidence of publication bias among included studies (B = -2.95, P-value = 0.271). Moreover, visual inspection of the funnel plot evidenced that symmetrical distribution of studies (Fig 3).

## Handling heterogeneity

Random effects model pooled estimate disclosed that there is significant heterogeneity. To handle this heterogeneity, sensitivity and sub-group analysis were performed. In sensitivity analysis, there were no studies that excessively influence the pooled prevalence of stunting (S1

**Table 1. Characteristics of included studies among children during their critical period in Ethiopia.**

| SN | Author/year | Study region | Study Design | Sample size | Prevalence of stunting | Quality Score (%) | Data collection technique | Fudging source |
|---|---|---|---|---|---|---|---|---|
| 1 | Sahiledengle et al, 2022 [21] | EDHS | Cross-sectional | 2146 | 27.1 | 87.5 | Anthropometry Questionnaire | Not funded |
| 2 | Tadele et al, 2022 [22] | SNNPRS | Cross-sectional | 362 | 21.82 | 75 | Interview and anthropometry measurements | Arbaminch University |
| 3 | Tafese et al, 2022 [28] | Oromia | Cross-sectional | 371 | 42.7 | 87.5 | Interview and anthropometry measurements | Hawassa University |
| 4 | Agedew and Chane,2015 [29] | SNNPRS | Cross-sectional | 567 | 18.7 | 75 | Interview and anthropometry measurements | not reported |
| 5 | Fekadu et al, 2015 [30] | Somalia | Cross-sectional | 214 | 22.9 | 75 | Interview and anthropometry measurements | not reported |
| 6 | Derso et al, 2017 [31] | Amhara | Cross-sectional | 587 | 58.1 | 75 | Interview and anthropometry measurements | University of Gondar |
| 7 | Berhe et al, 2019 [43] | Tigray | case control | 330 | NA | 90 | Interview and anthropometry measurements | not reported |
| 8 | Mulaw et al, 2020 [42] | Afar | case control | 381 | NA | 80 | Interview and anthropometry measurements | Mekelle and Samara university |
| 9 | Amera et al [33] | Amhara | Cross-sectional | 431 | 48.7 | 87.5 | Interview and anthropometry measurements | Not funded |
| 10 | Sewenet et al, 2022 [36] | Amhara | Cross-sectional | 421 | 36.8 | 87.5 | Interview and anthropometry measurements | Not reported |
| 11 | Yazew et al, 2021 [34] | Oromia | Cross-sectional | 500 | 27 | 87.5 | Interview and anthropometry measurements | Not reported |
| 12 | Worku et al, [35] | Adiss Ababa | Cross-sectional | 377 | 28.8 | 87.5 | Interview and anthropometry measurements | Not reported |
| 13 | Kidane et al, 2020 [4] | SNNPRS | Cross-sectional | 767 | 71.8 | 75 | Interview and anthropometry measurements | Not reported |
| 14 | Wolde et al, 2014 [32] | Oromia | Cross-sectional | 602 | 15.7 | 75 | Interview and anthropometry measurements | Not reported |

Fig). Sub-group analysis was done based on the region, residence and study year. The result of sub-group analysis based on region revealed that the highest prevalence of stunting was in Amhara region (47.91%). Whereas; the lowest prevalence was in Somalia regional state (22.90%) and the pooled prevalence of stunting also higher in rural residents (38.60%) (Table 2).

## Factors associated with stunting among children during their critical period

Child age, maternal education, maternal postnatal follow up, dietary diversity of the child (DD), maternal body mass index(BMI), feeding of Colostrum, exclusive breast feeding(EBF), number of under five children and wealth index of the household were significant factors for stunting(Table 3).

Five studies were included to assess the association between child age and stunting [4, 21, 28, 31, 42]. The pooled odds of developing stunting among children greater than 12 months were 3.83 times (OR = 3.83; 95% CI: 2.47–5.18) more likely than their counterparts (Fig 4); with statistically significant heterogeneity ($I^2$ = 97.76%, P-value<0.001). Egger's statistical test evidenced that there is no publication bias (P-value = 0.7061) and there was no single study that excessively affected the pooled estimate of stunting (S2 Fig).

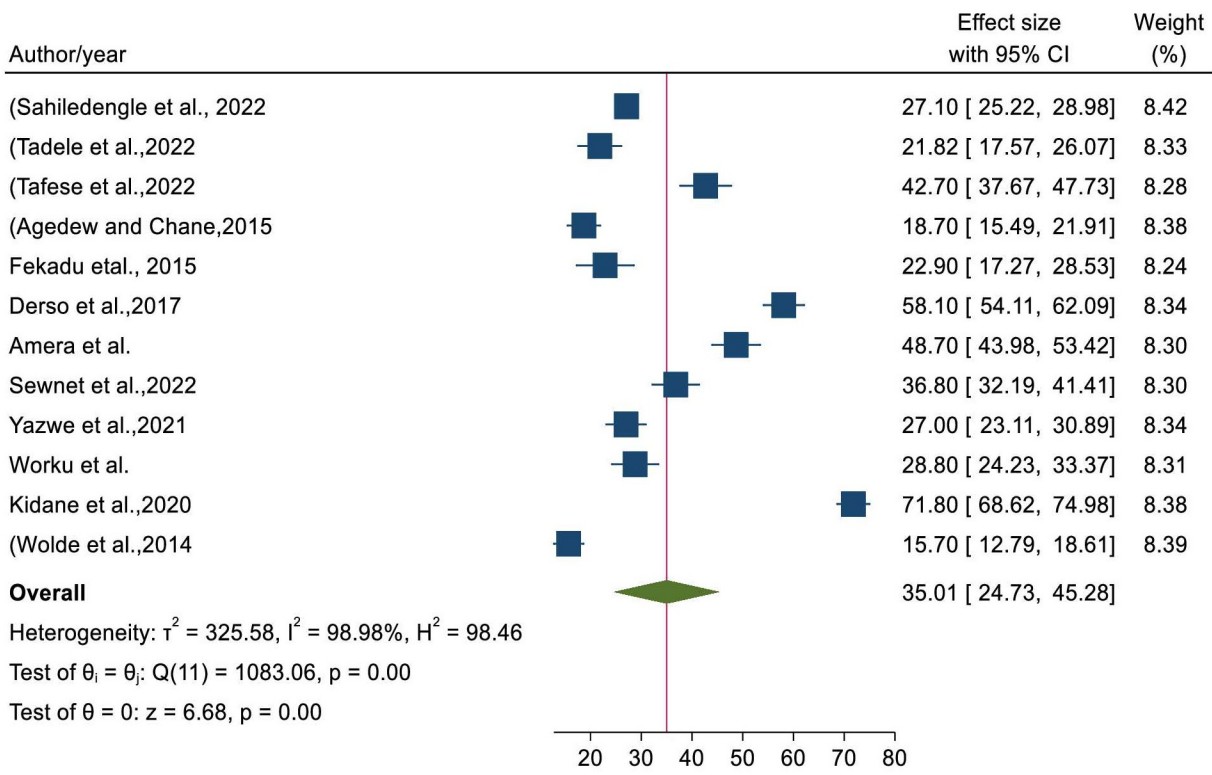

Fig 2. Pooled prevalence of stunting among children during critical period in Ethiopia.

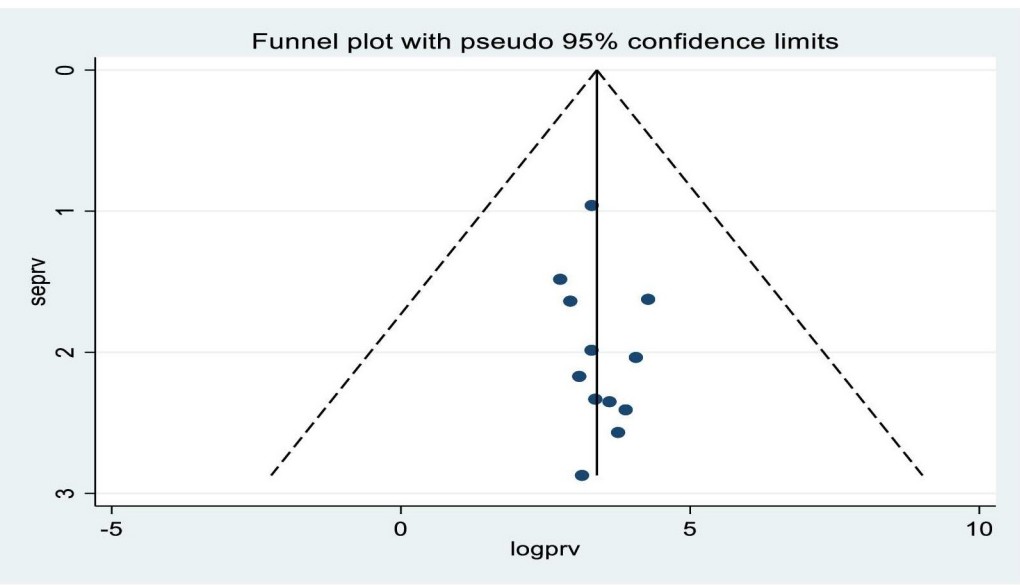

Fig 3. Funnel plot to test publication bias of the 12 studies.

**Table 2. Summary of sub-group analysis on prevalence of stunting among children during critical period in Ethiopia by region, residence and study year.**

| Variables | | Included studies | Prevalence (95%CI) | Heterogeneity (I²,p-value) |
|---|---|---|---|---|
| By region | Amhara | 3 | 47.91(35.52–60.29) | 95.74%,<0.001 |
| | Oromia | 3 | 28.35(13.70–42.99) | 97.69%,<0.001 |
| | SNNPRS | 3 | 37.45(1.71–73.19) | 99.68%,<0.001 |
| | Others | 3 | 26.84(24.51–29.16) | 24.29%, <0.001 |
| By residence | Rural | 8 | 38.60(22.68–54.52) | 99.21%,<0.001 |
| | Urban | 2 | 32.79(-24.95–40.63) | 82.87, <0.001 |
| | Urban& Rural | 2 | 23.00(14.78–31.23) | 94.90, <0.001 |
| By study year | ≥2020 | 5 | 40.41(20.38–60.45) | 99.11%,<0.001 |
| | <2020 | 7 | 31.13(20.85–41.41) | 98.49%,<0.001 |

The pooled estimates of four [4, 29, 32, 43] studies have determined the association between maternal education and childhood stunting. Statistical heterogeneity was observed among studies ($I^2$ = 89.73%, P-value<0.001) and there was no single study that excessively influenced the pooled effects of stunting (S3 Fig). Egger's statistical test showed that there is publication bias (P-value = 0.001). After trim and fill analysis (Fig 5), child from uneducated mothers had negative impact on stunting. Therefore, from trim and fill analysis the pooled odds of developing stunting among children with uneducated mothers were 2.90 times (OR = 2.90; 95%CI: 1.59–4.20) higher than children with educated mothers (Fig 6).

Two studies [4, 29] were included to determine the association between maternal post natal follow up and stunting. Children, whose mothers had no post natal follow up were 1.81 times more odds of developing stunting than their counterparts; with no statistical heterogeneity between studies(I = 0.0%, P-value<0.001).

A total of six studies [28, 30, 33, 34, 42, 43] were used to estimate the pooled effects of stunting and dietary diversity score of the child. From random effects model, the pooled estimates

**Table 3. Summary of the pooled effects of factors associated with stunting among children during their critical period in Ethiopia.**

| Variables | Category | OR (95%CI) | Heterogeneity(I2,P-value) | Egger's P-value | Total studies |
|---|---|---|---|---|---|
| Age of child | ≥12 month | 3.83(2.47–5.18) | 97.76,<0.001 | 0.7061 | 5 |
| | <12 month | 1 | | | |
| Maternal education | No | 2.90(1.59–4.20) | 89.73,<0.001 | 0.001 | 4 |
| | Yes | 1 | | | |
| Maternal PNC follow up | No | 1.81(1.51–2.10) | 0.00,<0.001 | 0.3240 | 2 |
| | Yes | 1 | | | |
| Dietary diversity score of the child | <4food diversity | 2.24(1.94–2.55) | 21.55,<0.001 | 0.0674 | 6 |
| | >4food diversity | 1 | | | |
| Maternal BMI | <18.5kg/m² | 3.22(2.39–4.04) | 40.58, <0.001 | 0.1945 | 2 |
| | >18.5kg/m² | 1 | | | |
| Feeding of Colostrum | No | 2.79(2.40–3.17) | 0.00,<0.001 | 0.7363 | 2 |
| | Yes | 1 | | | |
| Exclusive breast feeding | No | 2.65(1.60–3.71) | 83.92,<0.001 | 0.0968 | 2 |
| | Yes | 1 | | | |
| No. of under 5 children in the household | ≥2 | 2.76(2.27–3.26) | 0.00,<0.001 | 0.6388 | 2 |
| | <2 | 1 | | | |
| Wealth index | Poor | 2.23(1.89–2.58) | 0.00,<0.001 | 0.9347 | 2 |
| | Rich | 1 | | | |

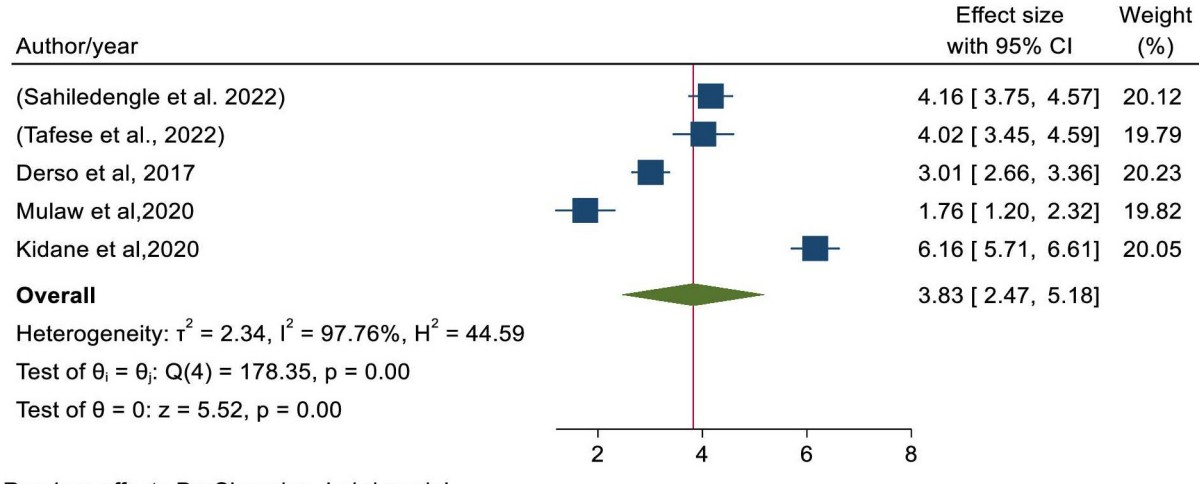

Fig 4. Forest plot showing the association between stunting and child age.

of stunting among children with less than food diversity score were 2.24 times odds of developing stunting(OR = 2.24; 95%CI:1.94–2.55)(Fig 7); with low heterogeneity($I^2$ = 21.55, P-value<0.001). Egger's statistical test revealed that there is no small study effect (P-value = 0.06674) and there was no single study that excessively affected the pooled estimate of stunting (S4 Fig).

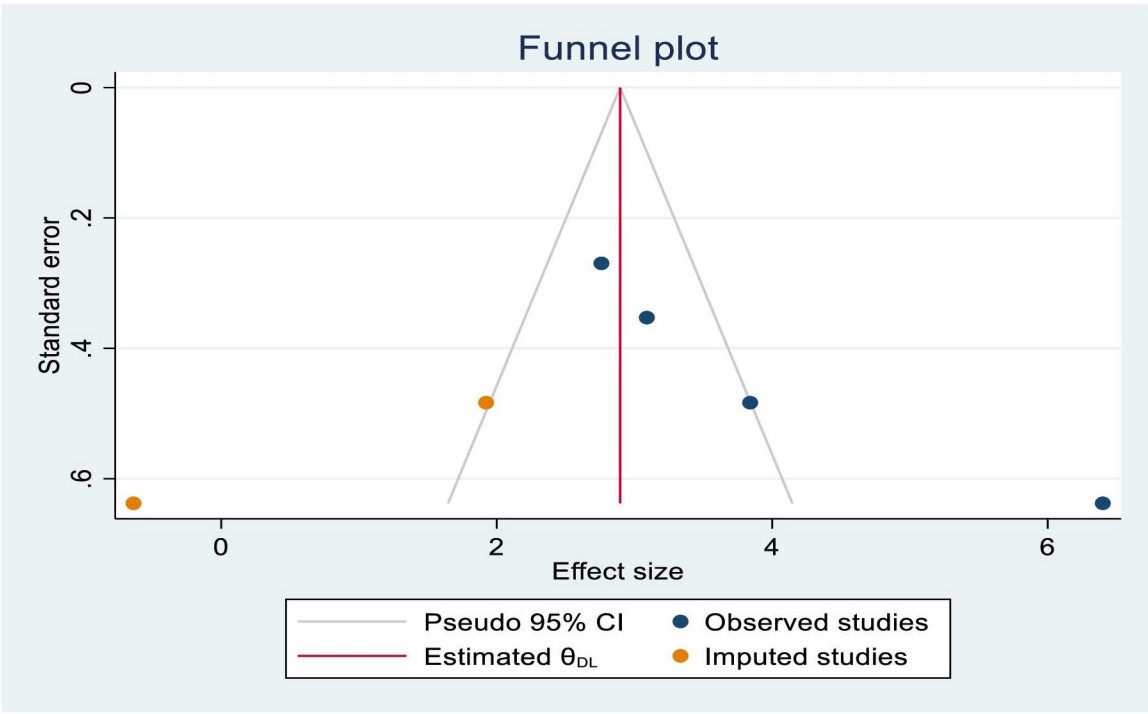

Fig 5. Trim and fill analysis funnel plot for maternal education status.

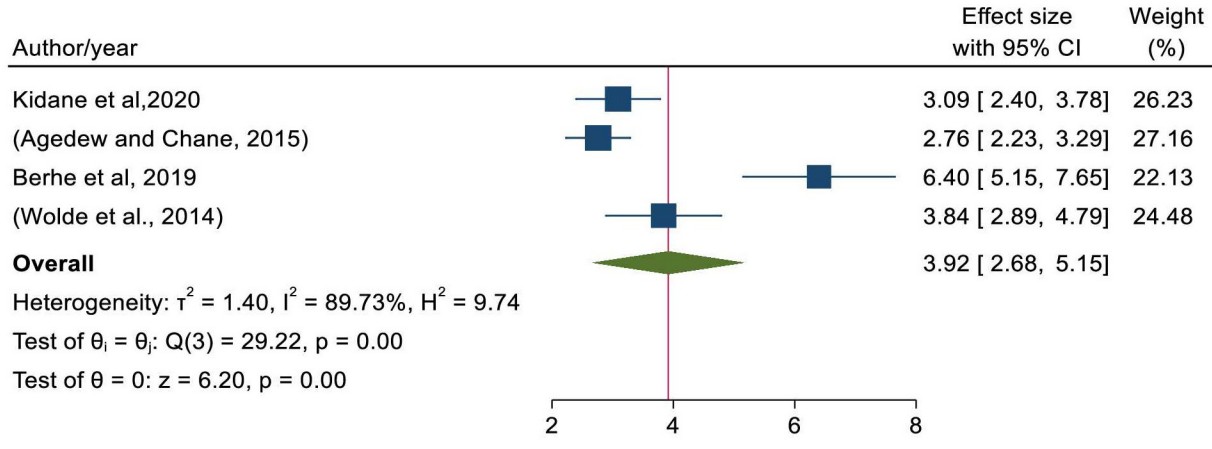

Fig 6. **Forest plot representing the association between stunting and maternal education status.**

As a result of 2 studies [42, 43], there is statistical significant association between maternal BMI and child stunting with low heterogeneity ($I^2$ = 40.58, P-value<0.001). Children whose mothers had less than 18.5 kg/m$^2$ is 3.22 times likelihood of developing stunting compared to their counterparts(OR = 3.22; 95%CI: 2.39–4.04).

The association between child stunting and feeding of colostrum was determined by two studies [4, 42]. The analysis disclosed that children who did not feed colostrum during neonatal period were 2.79 times more odds of developing stunting than their counterparts (OR = 2.79; 95%CI: 2.40–3.17); with no statistical heterogeneity between studies($I^2$ = 0.0%, P-value<0.001).

The analysis results of two studies [32, 42] revealed that the pooled effects of developing stunting among children with no exclusive breast feeding was 2.65 times (OR = 2.65; 95%CI:

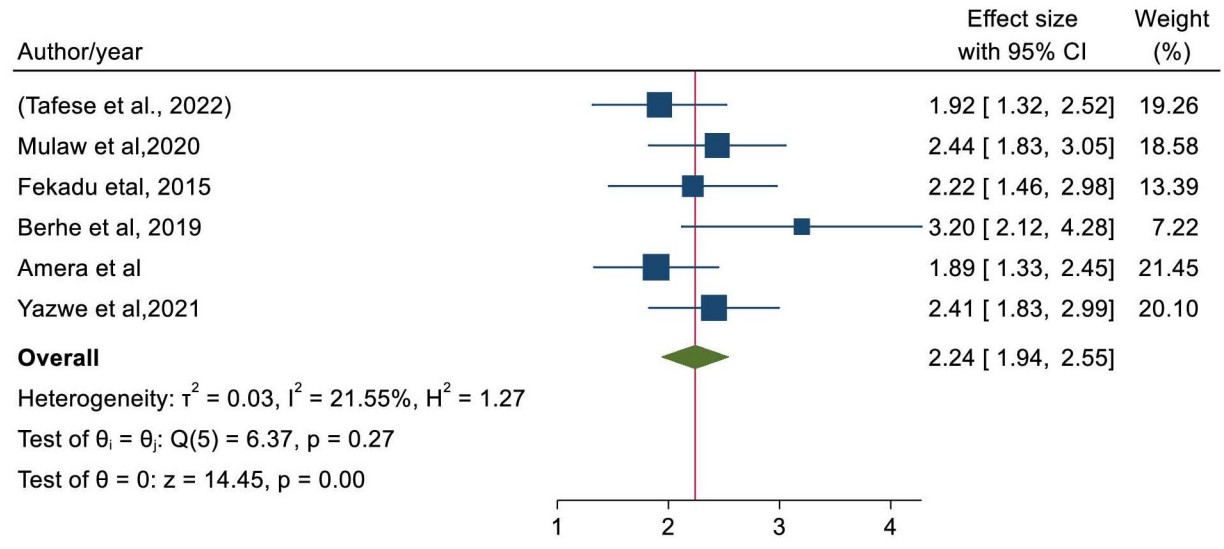

Fig 7. **Forest plot representing the association between stunting and child food diversity score.**

1.60–3.71) compared to children who exclusively breast feed. Furthermore, the associations of stunting and household wealth index /income were assessed by two articles [21, 31]. Children from poor household were more likely to be affected by stunting than children from rich family.

## Discussion

Stunting (height or length/age) is remaining a major public health problem among under five children in Ethiopia [27].

In this systematic review and meta-analysis, we sought to identify the pooled prevalence and associated factors of stunting in children under the age of 24 months and at their critical periods.

This comprehensive review and meta-analysis indicated a nationally pooled prevalence of stunting of 35.01 percent with a 95% confidence interval of 24.73 to 45.28, which was consistent with research carried out in Kenya [44], Zambia [14], Zimbabwe [45], Rwanda [17], and Tanzania [46]. However, the result of this study was lower than studies done in Malawi [15] and Burundi [16]. The pooled prevalence of stunting among children in this review was higher than studies investigated in China [47] and Ghana [48]. The possible reason could be due to differences in study setting, socio-demographic characteristics, time of study conducted and sample size. Moreover, in low and middle income countries, children are suffering from inadequate quantity, quality, and diversity of foods both due to lack of accessibility and awareness during their critical period of growth and development [5].

From sub-group analysis by region, the highest prevalence of stunting was observed in Amhara regional state (47.91%). The possible justification might be due to variation in study time and residence, sample size, study period and child feeding habits of the population [31]. Sub-group analysis by residence also disclosed that the prevalence of stunting was higher in rural residents (38.60%). This result was supported by a study performed in sub-Saharan Africa [49]. The plausible reason is that inadequate health services, food scarcity, and less awareness campaigns in rural areas. The other justification could be children in rural area are devastating nutritional status which is resulting from low quality care of mothers during perinatal period, delayed initiating of complementary feeding and low coverage of immunization of children. In contrary, children from urban area are beneficial in nutritional status over rural children interims of employment conditions and family networks to access better health care services [50].

In this study the pooled effects of child age greater than 12 month was 3.83 times higher than age less than 12months to develop stunting. This finding was in line with the study conducted in Pakistan [51, 52]. This is explained by children in the age range of 12–24 months are challenged with weaning effect that is related to the transition from breast-feeding to accustomed with other food items rather than mother's nutritious breast milk and lose passive immunity from their mothers which results stunting [18]. It could be also the extension of exclusive breast feeding which is not enough for growth and development beyond 6 months. Children, who are addicted for only breast milk, were faced difficulty to accept other food which exposed them for further stunting [17].

This review evidenced that maternal education has significant association with less than two years child stunting. This result was in agreement with the study conducted in Nepal and worldwide systematic review [53, 54]. Children from uneducated mothers were 2.90 times odds of stunting than their counterparts. This is justified by uneducated mothers tend to have low family income, spend less on proper nutrition and are more susceptible to growth failure due to lack of access to sufficient food of adequate quality and quantity [55]. To the opposite,

educated mothers have knowledge and awareness about child feeding practice, exclusive and timely imitation of complementary feeding [49]. In addition, maternal education can improve the household income which has direct impact on child nutrition and increase their health seeking behavior during illness [56].

The review also identified that maternal post natal follow up (PNC) has significant association with childhood stunting. Children whose mothers had no PNC follow up were more likely to be stunted than their counterparts. The possible reason could be due to mothers who followed postnatal care service have a chance to get advice regarding exclusive breast feeding, timely imitation complementary feeding and also they provided to give special compressive care for their child during their critical time from health care professionals [57].

From random effects model estimate, the pooled odds of developing stunting was higher among children who received less than four food score than children who provided greater than four varieties of food items which is similar to a study in Rwanda [19, 20]. If Children did not feed the recommended diversified food in their life window, it is inevitable to be exposed for stunted growth, development, morbidity and mortality. This is because, inaccessible to adequate diversified food that provide acceptable calories and micronutrients [58].

The odds of stunting were 3.22 times higher among children whose mothers were underweight than children from normal body mass index (BMI) mothers which was similar from study done in Tanzania and Ethiopia [59, 60]. This notion could be illustrated by the impact of maternal nutrition during pregnancy and lactation period, even before pregnancy since the process of becoming stunted typically begins in utero and mostly happened in the first 1000days [61]. Moreover, maternal undernutrition (stunting) restricts uterine blood flow, growth of the uterus and placenta which result in low birth weight, intrauterine growth restriction (IUGR) and growth restricted infants [62]. The study indicated that the pooled estimates of colostrum feeding and stunting were significantly associated. Children, who did not feed maternal colostrum during his first week of life, were more likely to be stunted compared to their counterparts. It is incongruent with the study done in Ethiopia and Zambia [14, 63]. This is reasoned by colostrum (first breast milk) which is produced in the first few days after delivery, is an unreserved first food for infants and considered as a first immunization against many bacteria and virus [64]. Therefore, avoidance of colostrum feeding during neonatal period imposed children for prelactal feeding and various infections which are also a risk factor for stunting [65, 66].

It was found that non-exclusive breast fed children were 2.65 times odds of stunting compared to their exclusive breast fed (EBF) counterparts. It was similar with the study investigated in Ghana, Sri Lanka and Ethiopia [67–69]. This is because, children who failed to attained EBF until six month lack essential nutrients from breast milk which is vital for growth and development, first immune against variety of infections, [70]. In addition, EBF used to protect against gastrointestinal infections that can cause nutritional depletion which in turn causes stunting. Failure of growth and development after birth is a reflection of EBF that is less precise and causes stunting [71].

The current review revealed that households with more than two under five children were more likely to be stunted than those who have less than two children. It is in line with the study in Ethiopia and Ghana [60, 72]. As a result, this is the fact that greater than two under five children in the household are overburden for the care giver to give the intended care. There could be also more competition and sharing of available foods that leads to food insecurity within the household, which in turn leads to stunting [42, 73].

Furthermore, family wealth status is a determinant factor for childhood stunting. Children from the poor family were two times more chance to develop stunting than from children rich family. This finding was similar from study in Ethiopia, Tanzania, Rwanda and Indonesia [59,

74–76]. Children from families with low-socioeconomic status have no purchasing power of the required food materials for their child and hence less likely to be exposed to good nutrition. Families' low-socioeconomic status also has negative effect on the household food access, utilization of health services, availability of improved water sources and sanitation facilities which further prone the child to stunted growth during their critical period [75].

This review followed considerable strength and limitations. It is the first study to assess the pooled prevalence of stunting and its determinants among children less than 2 years in Ethiopia. It adds an important and updated knowledge of stunting during their critical life period. Despite the strength of the study, it had the following limitations; only quantitative observational studies published in English language were included and significant heterogeneity was observed that might be undermining the pooled estimate of stunting. Hence, readers are requested to consider in using this study finding with these inherent limitations.

## Conclusion and recommendations

In conclusion, this review revealed that the pooled prevalence of stunting among children during their critical time is high and the highest prevalence was observed in Amhara region. Increased Child's age, no maternal education and no maternal postnatal follow up, less than four food diversity of the child, low maternal body mass index, failure to colostrum and exclusive breast feeding, two and more under five children in the household and poor wealth index of the family were determinants of stunting. Therefore, providing continuous maternal postnatal follow up, increase awareness of mothers on importance of colostrum and exclusive breast feeding, feeding of children the recommended variety of foods and at large to improve the wealth status of the households are crucial interventions to meet national and international target of zero stunting in children less than 2 years.

## Supporting information

**S1 Table. Studies search strategies and entry terms from different electronic data bases on the prevalence and determinants of stunting less than 24 months children in Ethiopia.** (DOCX)

**S1 Fig. Assessment of sensitivity analysis plot for prevalence of stunting among children during critical period.** (TIF)

**S2 Fig. Assessment of sensitivity analysis plot for factor child age.** (TIF)

**S3 Fig. Assessment of sensitivity analysis plot for the factor maternal education.** (TIF)

**S4 Fig. Assessment of sensitivity analysis plot for the factor child food diversity score.** (TIF)

**S1 Checklist. PRISMA checklist for included studies.** (DOCX)

**S2 Checklist. JBI critical appraisal checklist for included studies.** (DOCX)

## Author Contributions

**Conceptualization:** Amare Kassaw, Getachew Arage.

**Data curation:** Amare Kassaw, Yohannes Tesfahun Kassie, Shegaw Zeleke.

**Formal analysis:** Amare Kassaw, Amare simegn, Muluken Chanie Agimas.

**Funding acquisition:** Amare Kassaw.

**Investigation:** Amare Kassaw, Demewoz Kefale, Molla Azmeraw.

**Methodology:** Amare Kassaw, Worku Necho Asferi, Shegaw Zeleke.

**Project administration:** Amare Kassaw, Tigabu Munye, Solomon Demis, Amare simegn.

**Resources:** Amare Kassaw, Solomon Demis, Shegaw Zeleke.

**Software:** Amare Kassaw, Molla Azmeraw, Worku Necho Asferi, Shegaw Zeleke.

**Supervision:** Amare Kassaw, Yohannes Tesfahun Kassie.

**Validation:** Amare Kassaw, Yohannes Tesfahun Kassie.

**Visualization:** Amare Kassaw, Yohannes Tesfahun Kassie.

**Writing – original draft:** Amare Kassaw, Yohannes Tesfahun Kassie, Shegaw Zeleke.

**Writing – review & editing:** Amare Kassaw, Yohannes Tesfahun Kassie, Demewoz Kefale, Molla Azmeraw, Getachew Arage, Worku Necho Asferi, Tigabu Munye, Solomon Demis, Amare simegn, Muluken Chanie Agimas, Shegaw Zeleke.

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
