## [Decision Letter · Decision Letter 0]

23 Aug 2023

PONE-D-23-17150

Pooled prevalence and its determinants of stunting among children during their critical period in Ethiopia: A systematic review and Meta-analysis.

PLOS ONE

Dear Dr. Kassaw,

Thank you for submitting your manuscript to PLOS ONE. After careful consideration, we feel that it has merit but does not fully meet PLOS ONE’s publication criteria as it currently stands. Therefore, we invite you to submit a revised version of the manuscript that addresses the points raised during the review process.

We look forward to receiving your revised manuscript.

Kind regards,

Ayele Mamo Abebe, MSc in pediatric and child health nursing

Academic Editor

PLOS ONE

Journal Requirements:

https://www.dhsprogram.com/pubs/pdf/WP136/WP136.pdf

https://www.fantaproject.org/sites/default/files/resources/PROFILES-Brief-2-Stunting-Risk-Jun2018.pdf

https://univmed.org/ejurnal/index.php/medicina/article/view/879

https://www.researchgate.net/publication/359403221_PREVALENCE_OF_STUNTING_AND_ASSOCIATED_FACTORS_AMONG_EMPLOYED_AND_UNEMPLOYED_MOTHERS_OF_CHILDREN_AGED_6_TO_59_MONTHS_IN_DIRE_DAWA_ADMINISTRATION_EASTERN_ETHIOPIA_2021

file:///home/nkw-ld22-073/Downloads/SDG-briefing-note-3_nutritional-status.pdf

https://www.frontiersin.org/articles/10.3389/fpsyg.2022.847274/full

https://www.cell.com/heliyon/fulltext/S2405-8440(21)00226-7 _returnURL=https%3A%2F%2Flinkinghub.elsevier.com%2Fretrieve%2Fpii%2FS2405844021002267%3Fshowall%3Dtrue  

https://bmcpediatr.biomedcentral.com/articles/10.1186/s12887-017-0848-2

In your revision ensure you cite all your sources (including your own works), and quote or rephrase any duplicated text outside the methods section. Further consideration is dependent on these concerns being addressed.

Reviewers' comments:

Reviewer's Responses to Questions

**Comments to the Author**

1. Is the manuscript technically sound, and do the data support the conclusions?

Reviewer #1: Yes

2. Has the statistical analysis been performed appropriately and rigorously? 

Reviewer #1: Yes

3. Have the authors made all data underlying the findings in their manuscript fully available?

Reviewer #1: Yes

4. Is the manuscript presented in an intelligible fashion and written in standard English?

Reviewer #1: Yes

5. Review Comments to the Author

Reviewer #1: Manuscript Number: PONE-D-23-17150

Manuscript Title: Pooled prevalence and its determinants of stunting among children during their critical period in Ethiopia: A systematic review and Meta-analysis.

Congratulations dear authors on your scholarly work based on a priori protocol registered in PROSPERO; you have brought an important study problem with good findings that have public health importance in optimizing children’s health. However, there few methodological issues that I want you to address before considering the manuscript for publication.

General comment

There are several typological and grammar usage errors that need extensive proof reading for revisions.

Specific comments

Methods

I suggest the authors consider clear explanation of their PICO mnemonic to so that it will be easier for the readers. Just describe each component separately.

How did the authors look for grey literature?

Kindly append a table showing methodological quality of the appraised articles with the last column being ‘overall quality score’.

Explain if data transformation was required or undertaken when data were reported differently.

Results

Please, use PRISMA 2020 flow diagram. https://prisma-statement.org/PRISMAStatement/FlowDiagram

Please include two columns in Table 1: data collection technique (interview, observation, self administered questionnaire, etc) and funding source for each study (You can say not funded, not reported or name of funder if funded).

Good luck!!

6. PLOS authors have the option to publish the peer review history of their article (what does this mean?). If published, this will include your full peer review and any attached files.

Reviewer #1: **Yes: **Wubet Alebachew Bayih

---

## [Author Response · Author response to Decision Letter 0]

15 Oct 2023

# Editor Comment

1. Please ensure that your manuscript meets PLOS ONE's style requirements, including those for file naming

Authors’ Response 

We are grateful to this comment of technical relevance. Thus, we have ensured that our manuscript meets PLOS ONE's style requirements, including those for file naming.

2. We noticed you have some minor occurrence of overlapping text with the following previous publication(s), which needs to be addressed.

Authors’ Response 

Thank you. The authors have checked carefully the overlapping texts and paraphrased the duplicating works on the revised document. 

3. Please review your reference list to ensure that it is complete and correct. If you have cited papers that have been retracted, please include the rationale for doing so in the manuscript text, or remove these references and replace them with relevant current references. Any changes to the reference list should be mentioned in the rebuttal letter that accompanies your revised manuscript. If you need to cite a retracted article, indicate the article’s retracted status in the reference list and also include a citation and full reference for retraction notice. 

Authors’ Response

The authors are very grateful of these constructive comments. After read the reference lists carefully, we made a correction and addressed the incomplete one. The authors also have noticed the retracted articles on the reference list and removed these references and replace with relevant current references. But the authors did not removed the reference 39 and 40 since the source is book in which it has an updated version.

Reviewer one 

1. There are several typological and grammar usage errors that need extensive proof reading for revisions. 

Authors’ Response 

First of all, we thank you the reviewer for his constructive comments. After we have read carefully through the whole document, we properly addressed the concerned issues. 

Accepting the comment, the authors have read thoroughly and edited carefully the whole manuscript before submission. 

2. I suggest the authors consider clear explanation of their PICO mnemonic so that it will be easier for the readers. Just describe each component separately.

Authors’ Response 

 Thank you for your insight; accepting the comment, we made a clear explanation of PICO mnemonic on the revised version of the manuscript (page 5).

3. How did the authors look for grey literature?

Authors’ Response 

The authors have dealt on the issue of including grey literature. After deep discussion, we decided to include two articles based on the following reasons:

Based on the quality assessment, using critical appraisal checklists (JBI), these articles have low risk.

The authors consider that they had been assessed and evaluate since these articles were found from universities repositories.

Some evidences have recommended to consider these literatures if meeting certain criteria.

4. Kindly append a table showing methodological quality of the appraised articles with the last column being ‘overall quality score’.

Authors’ Response 

We accepted the comment and correct on the revised manuscript. Moreover, the methodological quality of the appraised articles were presented in detail in supplementary checklist two (S2 checklist) as cited on page 6.

5. Explain if data transformation was required or undertaken when data were reported differently. 

Authors’ Response 

Thank you for your concern. The authors did not undertake data transformation since it was not required. The data of all the included studies were presented similarly (log odds). 

6. Please, use PRISMA 2020 flow diagram

Authors’ Response 

Accepting the comment, the authors made a revision on the revised document. 

7. Please include two columns in Table 1: data collection technique (interview, observation, self-administered questionnaire, etc.) and funding source for each study (You can say not funded, not reported or name of funder if funded).

Authors’ Response 

Accepting the comment, we made a correction on the revised version of the manuscript (page 7-9).

---

## [Editor Report · Decision Letter 1]

7 Nov 2023

Pooled prevalence and its determinants of stunting among children during their critical period in Ethiopia: A systematic review and Meta-analysis.

PONE-D-23-17150R1

Dear Dr. Kassaw,

We’re pleased to inform you that your manuscript has been judged scientifically suitable for publication and will be formally accepted for publication once it meets all outstanding technical requirements.

Kind regards,

Ayele Mamo Abebe, MSc in pediatric and child health nursing

Academic Editor

PLOS ONE
---

## [Editor Report · Acceptance letter]

17 Nov 2023

PONE-D-23-17150R1 

Pooled prevalence and its determinants of stunting among children during their critical period in Ethiopia: A systematic review and Meta-analysis. 

Dear Dr. Kassaw:

I'm pleased to inform you that your manuscript has been deemed suitable for publication in PLOS ONE. Congratulations! Your manuscript is now with our production department. 

Kind regards, 

on behalf of

Assistant professor Ayele Mamo Abebe 

Academic Editor

PLOS ONE